# Handheld and Cost-Effective Fourier Lightfield Microscope

**DOI:** 10.3390/s22041459

**Published:** 2022-02-14

**Authors:** Laura Galdon, Hui Yun, Genaro Saavedra, Jorge Garcia-Sucerquia, Juan C. Barreiro, Manuel Martinez-Corral, Emilio Sanchez-Ortiga

**Affiliations:** 13D Imaging and Display Laboratory, Department of Optics, Universidad de Valencia, 46100 Burjassot, Spain; laura.galdon@uv.es (L.G.); hui.yun@uv.es (H.Y.); genaro.saavedra@uv.es (G.S.); jigarcia@unal.edu.co (J.G.-S.); juan.barreiro@uv.es (J.C.B.); manuel.martinez@uv.es (M.M.-C.); 2School of Physics, Universidad Nacional de Colombia, Medellin 050034, Colombia

**Keywords:** microscopy, light-field, 3D imaging

## Abstract

In this work, the design, building, and testing of the most portable, easy-to-build, robust, handheld, and cost-effective Fourier Lightfield Microscope (FLMic) to date is reported. The FLMic is built by means of a surveillance camera lens and additional off-the-shelf optical elements, resulting in a cost-effective FLMic exhibiting all the regular sought features in lightfield microscopy, such as refocusing and gathering 3D information of samples by means of a single-shot approach. The proposed FLMic features reduced dimensions and light weight, which, combined with its low cost, turn the presented FLMic into a strong candidate for in-field application where 3D imaging capabilities are pursued. The use of cost-effective optical elements has a relatively low impact on the optical performance, regarding the figures dictated by the theory, while its price can be at least 100 times lower than that of a regular FLMic. The system operability is tested in both bright-field and fluorescent modes by imaging a resolution target, a honeybee wing, and a knot of dyed cotton fibers.

## 1. Introduction

Fourier lightfield microscopy (FLMic) [1,2,3,4] is a reformulation of lightfield microscopy (LMic) [5,6,7,8,9,10,11] featuring the capacity of capturing directly, in a single shot, a collection of orthographic perspective images of 3D specimens. Due to the linear and spatially shift-invariant nature of captured views, FLMic is especially suited for easing the postprocessing and therefore for providing depth reconstructions with high and homogeneous resolution over a large depth of field [12,13,14,15]. Despite the short amount of time that has passed since FLMic was first reported [1], the number of applications for capturing dynamic biomedical images has increased significantly [16,17,18,19,20,21]. The FLMic can be built from scratch by aligning and adjusting many different elements, such as the illumination system, the sample holder, an infinity-corrected microscope objective (MO), the tube lens, relay lenses, the microlens array (MLA), and a digital camera. An attractive alternative is to build a Fourier lightfield accessory designed to be inserted at the camera port of a microscope [22]. In any case, the resulting FLMic can be somehow bulky, and with a final cost equal to several thousand dollars. Perhaps this fact has concealed the advantages of FLMic, and delayed the spread of the Fourier lightfield concept, which is potentially suited to application to image in-field sample volumes; for instance, living phytoplankton [22], zebrafish embryos [23], neuronal activity [24], and other potential applications such as microplastic screening, food processing chains, colloidal stability in material industries, amongst others.

With the aim of overcoming this bottleneck, in this paper we report the method of producing a compact, robust, reliable, and affordable FLMic. Thus, a hand-held and cost-effective FLMic was designed, built, and tested. In fact, we took profit of a new trend in microscopy: the design and production of low-cost miniscopes [25,26,27,28,29,30,31]. Following this trend an FLMic was designed with use of very affordable hardware without jeopardizing the optical performance. The resulting FLMic has a weight under the 400 g, reduced volume, and a cost that can be at least 100 times lower than that of a regular FLMic.

## 2. Materials and Methods

The FLMic design is based on two conjugation relations and on the key role played by two hard apertures. In summary, the MLA is virtually inserted, with the help of a telecentric optical relay, at the aperture stop of an infinity-corrected MO. This adjustment ensures that any microlens captures an orthographic perspective image of the 3D sample—usually named as the elemental image (EI). The number of microlenses that are fitted in the aperture stop of the MO equals the number of recorded EIs, from which the posterior digital processing allows the refocusing of the 3D sample volume. Additionally, a second hard stop is set in a plane between the lenses of the relay, so that it is conjugated with the object plane, and also with the pixelated sensor, which is set at the rear focal plane of the microlenses. The task of this second stop is to avoid overlapping between EIs, and therefore to fix the field of view (*FOV*) of the microscope. Thus, it is acting as the field stop.

The usual hardware setup makes the FLMic a somehow bulky device, the price of which can easily reach USD 1000, limiting its widespread use in terms of portability and cost. For these reasons, this works presents an easy-to-build, handheld, and cost-effective FLMic, whose 3D-render model is shown in Figure 1. In panel (a), the 3D model of the handheld and cost-effective lightfield microscope is shown. From this panel, the reader can see the physical dimensions that support the claim of handheld; the illumination system and the cover slide manipulation stage are included in the drawing. In panel (b), an exploded drawing where the components of the microscope and their assembly are presented. The LED is the main component of the illumination system; the field stop is illustrated in the alignment stage and the sample is also drawn with its corresponding manipulation stage.

As a proposal to build a compact and cost-effective FLMic, we focused on two main points: (i) the use of imaging lenses with optical performance roughly similar to that of an infinity-corrected MO, but some orders of magnitude less costly; and (ii) the determination of a method of removing the relay lenses and the corresponding field stop. To accomplish the point (i) we chose, as strong candidates, the M12 lenses typically used in surveillance cameras. These lenses, which have a short focal length, were originally designed for providing aberration-compensated images of far and large scenes. Interestingly, this means that by simply inverting the light-rays trajectory, the M12 lenses can be used for producing at the infinity the aberration-compensated image of a microscopic sample, provided it is placed at the M12 focal plane. In this sense, the inverted M12 lenses behaves in the same way as an infinity-corrected MO. Thanks to their mass production, these lenses have a cost that is at least two-orders of magnitude lower than an optically equivalent infinity-corrected MO [32]. Naturally, the main concern for using this configuration is the presence of optical aberrations in the image plane that could ruin the performance of the proposed FLMic. However, as it will be discussed in Section 3, the aberration impact is low, allowing the overall performance of the proposed FLMic to be quite competitive.

The challenge of point (ii) is twofold. It is necessary to find a method of placing the MLA at the exit pupil of the M12 lens and also a method of inserting a field stop to prevent the overlapping between EIs. Despite the fact that apparently there is no available technical information on the M12 optical design, we found experimentally that the exit pupil of said lens is close to its outermost surface. This means that placing the by MLA in close contact with the M12 lens, the former is in the proximity of the exit pupil. The consequence of not locating the MLA exactly at the MO exit pupil is not a reduction in the optical quality of the captured EIs, but the introduction of some vignetting in the outermost EIs. The field-stop problem is solved here by placing a diaphragm just at the sample plane. In other words, only the required *FOV* is illuminated. This hard stop behaves indeed as the effective entrance window, which seen from the image plane is therefore the field stop of the FLMic. Thus, its size must be chosen in such a way that the *FOV* of the FLMic is maximized, namely, allowing the EIs to be tangent to each other. It is important to place this diaphragm in close contact to the sample to ensure that both are in focus, that is, they lay within the depth of field (*DOF)*.

The choice of the digital camera must also follow the cost-effective philosophy of the proposed FLMic. Fortunately, the mass production for diverse purposes has made available very competitive digital cameras with prices that are very accessible. Because of its great versatility and attractive tradeoff between price and performance, in this work the HQ Raspberry-Pi Camera was selected.

In overall, while the replacement of the infinity-corrected MO by the M12 lens guarantees the cost-effectiveness of the system, the elimination of the relay lens and the relocation of the field stop secures the compactness and portability of the resulting microscope, which articulated with the HQ Raspberry-Pi Camera turns the designed FLMic into a very competitive device, as shown in the following sections. Raspberry Pi can be powered by regular off-the-shelf batteries. It needs a power supply of voltage 5.1 V while the recommended capacity depends on the model. Depending on the model, it can be powered via Micro USB or USB-C. Therefore, having the right connector, a power bank or a solar cell can be used to supply the Raspberry Pi. To visualize the captured EIs and reconstructed images, a HDMI monitor is also necessary or the capabilities of the Raspberry Pi can even be used to send information to a mobile phone, to use it as a display.

### 2.1. Design Parameters

The design and build of the FLMic illustrated in Figure 1 followed the equation and process of design formerly reported in [2,4]. In these references, all the details and governing equations can be found. However, they are adapted here to the new situation in which the relay is not used.

The first parameter to control is the spatial resolution of the handheld FLMic. The lateral resolution limit of the directly captured EIs, strictly understood as the capability of resolving two equally emitting incoherent light points separated a distance $\rho_{EI}$ in the object space, is given by sum of the diffractive $\rho_{DIF}$ and geometric $\rho_{GEO}$ factors. It is given by:(1)$$\rho_{EI}=\rho_{DIF}+\rho_{GEO}=\lambda \frac{N}{2NA}+2\frac{\delta }{M_{T}}.$$

In Equation (1), *N* is the number of microlenses fitted within the aperture stop of the MO, λ the illuminating wavelength, and *NA* the numerical aperture of the MO. Besides,
(2)$$M_{T}=\frac{f_{MLA}}{f_{MO}},$$
is the total magnification of the FLMic, being $f_{MLA}$ and $f_{MO}$ the focal length of the MLA and the M12 lens, respectively. Finally, δ is the pixel pitch of the digital camera.

For the selected M12 lenses *NA* = 0.2, hence, a small angle approximation can be utilized to compute the number *N*:(3)$$N\approx 2\frac{NAf_{MO}}{p},$$
with *p* the pitch of the MLA. The value of *N* must be chosen to be between 3 and 5 to have enough EIs to compute the refocused images while the spatial resolution is not too penalized, even though other choices of *N* are feasible if required.

To ensure that the EIs are tangent to each other over the digital sensor, the size of the image of the field stop must match the value of *p*. As for the proposed handheld FLMic the field stop is in contact with the sample plane, the reachable *FOV* is equal to the field-stop diameter, $\phi_{FS}$, which is then given by:(4)$$FOV=\phi_{FS}=\frac{p}{M_{T}}.$$

Finally, the *DOF* of the proposed handheld lightfield microscope is also the result of the entanglement of the diffractive and the geometrical factors. Accordingly, the *DOF* yields:(5)$$DOF=2\lambda \frac{N^{2}}{NA^{2}}+\frac{N}{NA}\frac{\delta }{M_{T}}.$$

### 2.2. Design Workflow

The use of the M12 lens as infinity-corrected MO, dictates the working *NA* = 0.2 reported by the manufacturer, and a set of focal lengths ranging from 0.76 to 20 mm [32]. Additionally, the use of blue-light illumination enhances the performance of the microscope in terms of the spatial resolution, hence, a *λ* = 490 nm was chosen. As mentioned, a HQ Raspberry-Pi Camera (square pixels of δ=1.55 µm) was selected. The set of selected parameters (*NA*, *λ*, and δ) are those that the designer must keep fixed to pursue the idea of cost-effectiveness, while the optical performance is not too jeopardized. The focal lengths of the M12 and MLA and the pitch of the latter are the variable parameters to be examined to look for the optimal handheld FLMic. We selected, as well, a pitch p=1 mm for the MLA, which is commercially available with many different values of $f_{MLA}$. Hence, using as a guiding parameter in the design to reach the best possible spatial resolution, *N* was constrained to be between 3 and 5, what leads to 7.5 mm $<f_{MO}<12.5$ mm. Without lack of generality in this design, $f_{MO}=8$ mm was chosen, which is available in the ArduCam^®^ catalog. For this value, a spatial resolution limit of 8.0 µm is possible for $f_{MLA}>6.0$ mm. The choosing of the value of $f_{MLA}$ can be led by what feature of FLMic wants to be prioritized. For example, the *FOV* is about 1.2 mm if $f_{MLA}=6.0$ mm, though one may choose a slightly lower *FOV*, for instance 1 mm (what is reached as for $f_{MLA}$ = 8 mm), to obtain a larger *DOF*.

In summary, a FLMic expressing as sought features the compactness and cost-effectiveness was designed. The use of cost-effective hardware elements and the elimination of the optical relay might have an impact on the image quality. Especially, the optical performance of the M12 lens, originally designed for surveillance cameras, utilized in the proposed FLMic as infinity-corrected MO, might introduce doubts. The overall optical performance of the handheld and cost-effective FLMic is evaluated in the Section 3; special emphasis is focused on the spatial resolution of the microscope. Once the optical performance was evaluated, the handheld and compact FLMic was utilized to image biological samples, operating in bright-field mode, and an intricate knot of fibers, operating in fluorescent mode.

### 2.3. Reconstruction Algorithms

FLMic has the capability to capture the 3D information of thick samples by means of an array of microlenses, each of which provides a different orthographic perspective of the scene. These EIs contain different angular information, which is the basis of the refocusing algorithms. The standard back-propagation shift and sum (S&S) algorithm [3] is based on summing the EIs stacked and shifted with respect to the central one. The obtained depth images depend on the number of shifted pixels, *n*, through
(6)$$z=n\frac{f_{MO}^{2}}{f_{MLA}}\frac{\delta }{p},$$
z being the depth coordinate as measured from the M12 front focal plane.

In the case of sparse fluorescent samples, instead of applying the classic S&S algorithm, one can take advantage of the shift and multiply (S&M) method [12], which provides optical sectioning capability, namely, it can remove light from out-of-focus planes and compute sectioned images at different depths of the sparse sample. This algorithm is also a back-propagation algorithm, so the axial distance between refocused planes continues to be that of Equation (6).

Even though there are other different reconstruction algorithms, based on the deconvolution with a synthetic impulse response [7,14], that can be applied to a Fourier lightfield imaging, in this work easy to implement algorithms with high throughput were utilized. Thus, we used the S&S algorithm in the case of bright-field mode and the S&M algorithm for fluorescence images.

## 3. Results

In Figure 2, a photograph of the handheld and cost-effective FLMic is shown. The microscope was built with an M12 lens (NA=0.2 and $f_{MO}=8.0$ mm), manufactured by ArduCam^®^. The chosen MLA has a focal length of $f_{\mathrm{MLA}}=7.94$ mm and pitch of p=1.0 mm (model APH-Q-P1000-R3.63, manufactured by AMUS^®^). As a digital sensor, an HQ Raspberry-Pi Camera with 3648×2736 square pixels of δ=1.55 μm, was selected. For illuminating the sample an off-the-shelf ultra-bright blue LED, with emission peak wavelength of 490 nm and a full width at half maximum (FWHM) of 40 nm, was used. The picture also includes the Raspberry-Pi utilized to control the digital camera and the illuminating LED.

To limit the *FOV* and avoid the overlapping between EIs a field stop of 1 mm in diameter was placed close to the sample. This stop was built by drilling a hole with the said size on an aluminum plate foil with 0.8 mm thickness. This maintains the cost effectiveness of the microscope. In the case of fluorescence, to collect the light emitted by the sample and to block the remaining light, we used recycled color plastic filters extracted from 3D anaglyph glasses. Again, this retains the system’s cost-effective design. Despite using low-cost elements, their performance is strong enough. The spectrum of the plastic filter was measured, obtaining a transmission peak at 610 nm and a FWHM of 60 nm. The weight of the microscope, below 400 g, can be further reduced by using specially designed and built (even by a 3D printer) hardware; however, in the prototype off-the-shelf elements were utilized. For this microscope, the expected theoretical values are N=3.2, $\rho_{EI}=7.1$ μm, FOV=1.0 mm, and DOF=276 μm.

To help the reader visualize the handheld nature of the designed microscope, in Appendix A a video recording of a microscopist operating the handheld FLMic is shown.

To show the effectiveness of the handheld and cost-effective FLMic, several experiments were completed. In the first experiment, the spatial lateral resolution of the microscope was studied by imaging a negative hi-res USAF 1951 test target. Figure 3 shows, in the upper row from left-to-right: the complete set of the recorded EIs, the zoomed-in central EI, and a zoomed-in area including the elements from 1 to 4 of group 7. The set of recorded Els shows the 3,2 EIs (along the horizontal direction) for which the lightfield microscope was designed. From the zoomed-in area of the elements of group 7, an intensity profile along the green line is plotted. Here, the reader observes that the microscope can resolve with a contrast of 10% the element 1 of group 7, namely, it has a spatial resolution limit of 7.82 µm (or equivalently a cut-off frequency of 128.0 lp/mm). The lower row of Figure 3 shows the same resolution study for the reconstructed image at the best focus plane. As it can be observed, both the reconstructed image and the EIs show the same resolution. From the same set of EIs images shown in Figure 3, the *FOV* of the microscope was measured to be 0.95 mm in diameter. Note that these two values are in very good accordance with the expected theoretical ones.

To measure the *DOF*, the USAF 1951 test target was imaged while it was axially displaced backwards and forwards. The *DOF* is the distance the test is displaced in both directions till the resolution is reduced by factor $\sqrt{2}$, i.e., in three elements with respect to the best focus plane. Taking this criterion, a *DOF* = 280 µm was measured for the handheld and cost-effective FLMic, which again matches very well with the expected value. The very good agreement between expected and experimental values validates the assumption that the impact of aberrations, due to the use of M12 lenses, is negligible.

Once the optical performance of the FLMic was evaluated, its feasibility to capture the three-dimensional information of thick samples was tested, both in bright-field and fluorescence imaging. As a bright-field sample, a honeybee wing was utilized. The captured EIs are shown in Figure 4a. Using the seven non-vignetted EIs, nine refocused planes were computed by means of the S&S algorithm, which are presented in Figure 4b. The axial step between the refocused images for this configuration is 12.5 µm. In Appendix A we show a movie in which the frames are refocused images.

The handheld FLMic microscope can be also applied to obtain images of fluorescent samples. For the next experiment, a sample of a knot of cotton fibers, dyed with Rhodamine 123, was utilized. The set of recorded EIs is shown in Figure 5 panel (a). Instead of applying the classic S&S algorithm, the shift and multiply (S&M) algorithm was utilized to take advantage of its optical sectioning capability in the case of sparse fluorescent samples. In panel (b), several refocused images are presented. A 3D rendering, obtained by applying to these depth planes a maximum intensity projection (MIP) algorithm, was also created. The 3D rendering is composed of 46 refocused planes, so the rendered volume has a size of 950 × 950 µm^2^ in the transverse axes and 575 µm in the axial one in the object space. In Appendix A, different views of the said rendering are shown. The voxel size of the rendering is 1.89 × 1.89 × 6.25 µm^3^, where the larger side corresponds to the axial direction.

## 4. Conclusions

The design, implementation, and testing of the most portable, easy-to-build, robust, handheld, and cost-effective Fourier Lightfield Microscope (FLMic) to date, were presented. The design of the presented FLMic is an alteration of the original one with the removal of the relay lenses, the relocation of a cost-effective handmade field stop, and the replacement of the expensive infinity-corrected microscope objective (MO) by a surveillance M12 lens. This latter cost-effective lens, as utilized in the reverse direction of light gathering of its meant design, behaves as a competitive infinity-corrected MO with NA=0.2 costing at least 100 times less than an equivalent regular commercial MO. In this system, the EIs are formed at transverse positions close to the optical axis, thus, the possible aberrations introduced by this non-scientific-grade lens are not relevant in this region, resulting in images of high optical quality. The design also accomplished the proper matching of the parameters of the MLA with the chosen digital camera to not jeopardize the optical performance of the complete FLMic. For the implementation of the proposed microscope, a M12 lens with 8 mm focal length was housed in a home-made cage that also allocates the MLA in contact with the lens. At the focal plane of the MLA an HQ Raspberry-Pi Camera was placed to record the elemental images. These images were processed via the regular refocusing algorithm to produce the refocused images at different depth planes. The proposed FLMic showed the optical performance dictated by the theory, similarly to that of a bulkier and much more expensive FLMic built based on expensive MO and relay lenses. The results showed that despite its at least 100 times lower price, optically-wise the proposed FLMic exhibits a competitive performance, while maintaining the real-time single shot 3D imaging capabilities. Therefore, the FLMic proposed here shows great promise as a handheld device for in-field 3D-imaging studies where a cost-effective device with competitive optical performance is the preferred choice.

## Figures and Tables

**Figure 1 sensors-22-01459-f001:**
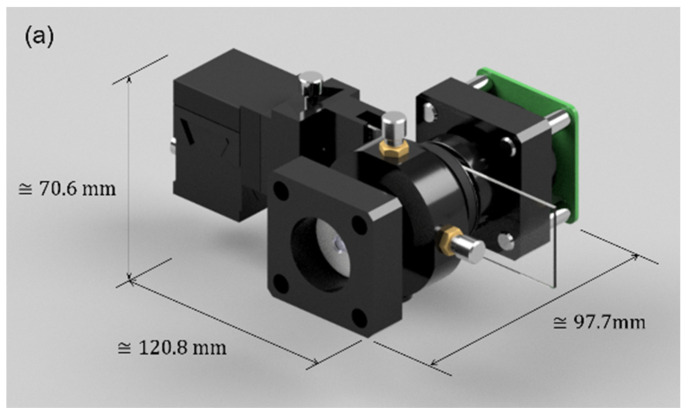
Scheme of the proposed hand-held and cost-effective FLMic. (**a**) The 3D modeling of the microscope. (**b**) Exploded drawing of the microscope.

**Figure 2 sensors-22-01459-f002:**
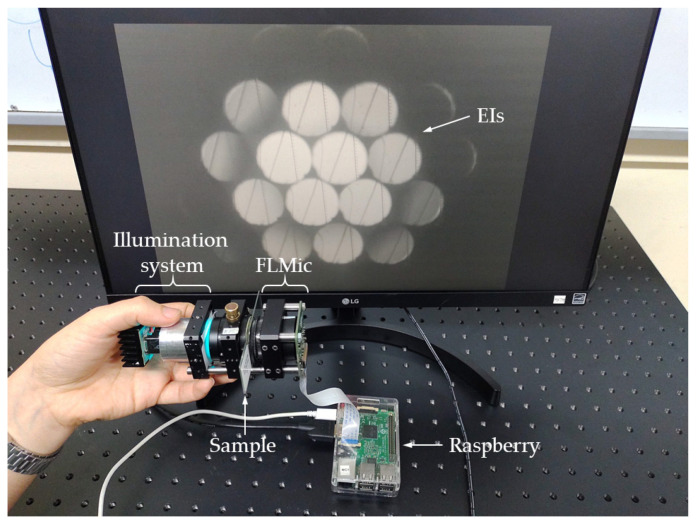
Photograph of the handheld and cost-effective FLMic.

**Figure 3 sensors-22-01459-f003:**
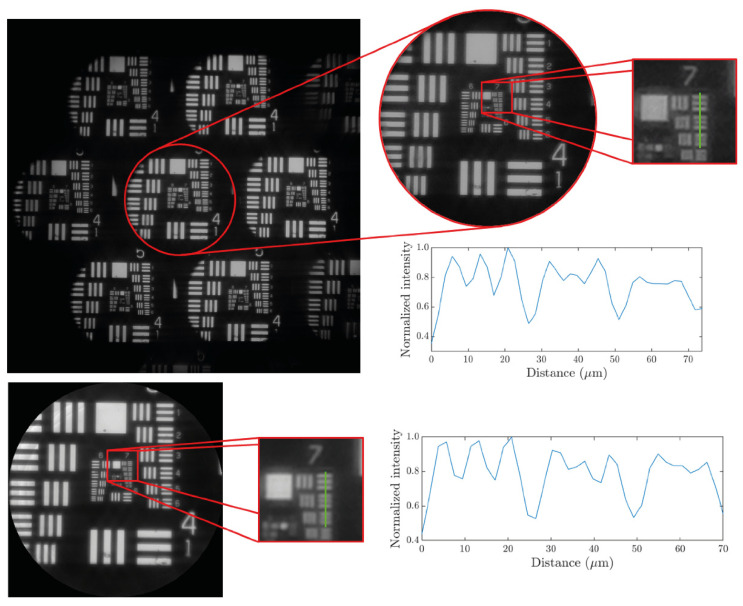
Study of the spatial resolution of the handheld and cost-effective FLMic. In the upper row, the lateral resolution of the central image is studied and in the lower row that of the refocused image at the best focus plane.

**Figure 4 sensors-22-01459-f004:**
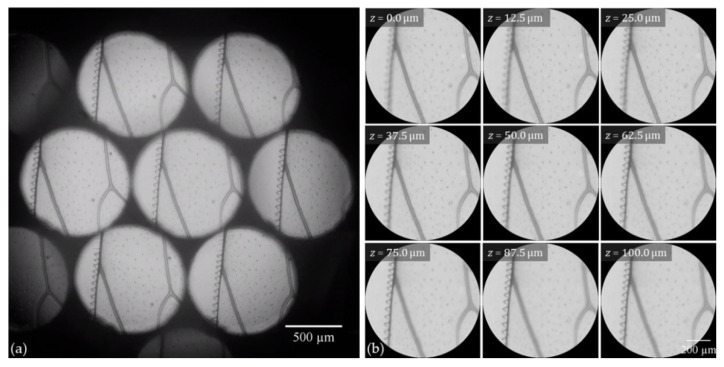
(**a**) Elemental images of a honeybee wing captured using the handheld and cost-effective FLMic; (**b**) refocused images at different depth planes. The scale bar indicates the equivalent distance in the object space.

**Figure 5 sensors-22-01459-f005:**
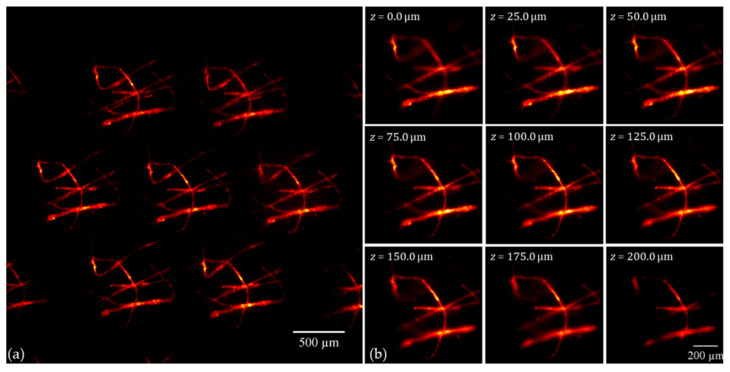
(**a**) EIs of the fluorescence knot of cotton fibers obtained with the handheld and cost-effective FLMic; the scale bar indicates the distance in the object space; (**b**) refocused images at different depth planes.

## Data Availability

The data presented in this study are contained within the article and the Appendix A.

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
