# Peer review of "Handheld and Cost-Effective Fourier Lightfield Microscope"

_sensors, 2022, doi:10.3390/s22041459_

Round 1

Reviewer 1 Report

Dear authors,

Thank you for submitting this manuscript. In this manuscript, you show the concept, theoretical derivation, construction and test of a cost-effective lightfield microscope. The motivation is clear, and the paper structure is well organized. I recommend this manuscript be published. Here are some comments:

  1. From line 48 to 59, the authors describe the optical stacking of the microscope. I recommend adding a figure "schematic optical setup" to help readers quickly understand your design. Or you could move Figure 1b to line 59.
  2. Your motivation is to build a compact and low-cost system, so it is understandable to sacrifice some optical performance. Is it possible to estimate system errors? For example, line 79 to 93 describe how you roughly locate the position of key elements. How does position error influence the equations in section 3.1 and 3.3?

Author Response

R) Dear authors,

Thank you for submitting this manuscript. In this manuscript, you show the concept, theoretical derivation, construction and test of a cost-effective lightfield microscope. The motivation is clear, and the paper structure is well organized. I recommend this manuscript be published. Here are some comments:

A) We would like to thank the reviewer for his/her comments that helped us to improve the manuscript quality.

R) 1. From line 48 to 59, the authors describe the optical stacking of the microscope. I recommend adding a figure "schematic optical setup" to help readers quickly understand your design. Or you could move Figure 1b to line 59.

A) We agree with the reviewer’s comment. We have moved the entire Figure 1 to the beginning of Section 2.

R) 2. Your motivation is to build a compact and low-cost system, so it is understandable to sacrifice some optical performance. Is it possible to estimate system errors? For example, line 79 to 93 describe how you roughly locate the position of key elements. How does position error influence the equations in section 3.1 and 3.3?

A) The facts of not locating the microlens array exactly at the objective exit pupil or not placing the field stop at the object plane, do not imply a reduction of the optical performance of the captured EIs. Therefore, resolution and depth of field are not affected, as demonstrated by results of section 3. The noticeable flaw of the axial misplacement of both elements is vignetting (as stated in the previous version of the manuscript) in the outermost EIs, which could reduce the number of useful elemental images for the reconstruction algorithm. As for the proposed and implemented setup, this vignetting effect is negligible.

Reviewer 2 Report

See attached file

Author Response

R) This letter reports the design, construction, and testing of a miniaturized Fourier light-field microscope (FLMic) suitable for field 3D microscopy of sample volumes on the order of 1mm3. Design modifications are justified to lower the size and cost relative to a laboratory FLMic, specifically the FLMic demonstrated by the same group in Ref. 2, and 3D micrographs are shown to qualitatively demonstrate the performance. While the claim that design compromises and low-cost optical components cause only “a relatively low impact in the optical performance” is not quantified, this is probably fine for prospective users who need only qualitative field microscopy. It would be worthwhile, in a subsequent paper, to compare this FLMic with the laboratory FLMic to quantify the effects of design compromises and low-cost components, for instance the inexact placement of the field stop. Overall the paper is sufficiently novel, important, and technically correct to be published with only the following minor changes:

A) First, we would like to thank the reviewer for his/her comments that helped us to improve the manuscript quality. Moreover, we will consider his/her suggestions about possible additional projects.

R) 1. The Introduction should mention potential applications, like field microbiology, microplastic sampling, etc. rather than just “a large variety of bioimage types.” Likewise the abstract should describe the test specimens rather than just “some bio- and material-samples.”

A) We thank the reviewer for his/her comment. We have added the following sentences and new references:

“… imaging a resolution target, a honeybee wing and a knot of dyed cotton fibers.”

“… which is potentially suited to be applied to image in-field sample volumes; for instance living phytoplankton [a], zebrafish embryos [b], neuronal activity [c] and other potential applications like microplastic screening, food processing chains, colloidal stability in material industries, amongst others.”

[a] N. Incardona, A. Tolosa, G. Scrofani, M. Martinez-Corral. and G. Saavedra, “The lightfield microscope eyepiece,” Sensors 21, 6619 (2021).

[b] G. Scrofani, G. Saavedra, M. Martínez-Corral, E. Sánchez-Ortiga, “Three-dimensional real-time darkfield imaging through Fourier lightfield microscopy,” Opt. Express 28, 30513-30519 (2020)

[c] R. Prevedel, YG. Yoon, M. Hoffmann, N. Pak, G. Wetzstein, S. Kato, T. Schrödel, R. Raskar, M. Zimmer, E. S. Boyden, “Simultaneous whole-animal 3D imaging of neuronal activity using light-field microscopy. Nat Methods 11, 727–730 (2014).

R) 2. The quoted specifications should include power requirements. Can the FLMic be powered by an off-the-shelf portable battery or solar cell?

A) We would like to thank the reviewer’s suggestion. We added the following information:

“Raspberry Pi can be powered by regular off-the-shelf batteries, it needs a power supply of voltage 5.1 V while the recommended capacity depends on the model. Depending on the model, it can be powered via Micro USB or USB-C. Therefore, having the right connector, a power bank or a solar cell can be used to supply the Raspberry Pi. To visualize the captured EIs and reconstructed images, a HDMI monitor is also necessary or even one can use the capabilities of the Raspberry Pi to send information to a mobile phone, to use it as a display.”

R) 3. I think the resolution at Eq. 1 should be, as stated in Ref. 2, the larger of the diffraction-limited or pixel-limited resolution, not the sum.

A) Considering the maximum value between the two resolution limits is an approximation. If the optical system is strongly diffraction limited, the effect of the pixel size is negligible, so it can be approximated that the resolution of the system is given by the diffractive resolution limit. The opposite applies if the resolution limit dictated by the pixel size is much larger. In the case of FLMic, both resolution limits are close in value, so neglecting one of them over the other is not a reliable approximation.

To be more precise, the final image provided by the optical system is the convolution between the optical transfer function (OTF) of the sensor and the diffraction OTF. The cut-off frequency of the resulting function is the resolution limit of the system. Hence, both the diffraction-limited and the pixel-limited resolutions must be added, especially if both values are close to each other. For this reason we decided to keep the resolution limit as the sum of both limits.

R) 4. On line 259 the rendered volume should be in um2 not mm2.

A) Thanks for the correction. We have changed it.

R) 5. Correct the following typographical and language errors:

Abstract line 11 should read “off-the-shelf”

The sentence on lines 14-16 of the Abstract is awkward and should be reworded, to something like “…reduced dimensions and light weight, which, combined with its low cost, make the FLMic a strong candidate…3D imaging capabilities are needed.”

“Low impact” should not be hyphenated on line 17.

Misspelling on line 119

A) We have made the corresponding changes.

Reviewer 3 Report

In this manuscript, the authors presented a compact, handheld, and cost-effective Fourier light field microscope (FLMic) for bio-sample imaging in both bright-field and fluorescent modes. The authors employed a M21 lens to replace infinity-corrected microscope objective, and used a field stop in illumination system to simplify the imaging system. Both the theoretical analysis and the experiment results have been given to demonstrate the performance of the proposed device. I think the proposed methods could be helpful in the applications of light field microscopy. Therefore, I suggest that the manuscript would be accepted if the authors make some modifications. Here are my comments:
1.In page 6, the authors discussed the spatial lateral resolution of the imaging system by imaging a USAF 1951 test target. They calculated the spatial lateral resolution by using the elemental images rather than the refocused images. I suggest the authors use the refocused images to calculate the spatial lateral resolution as the experiment results.
2.In page 8, the authors demonstrated the fluorescent imaging results. I suggest the authors provide some further details about experimental process. For example, did the authors use the optical filter? If so, how did the author mount the filter in the proposed system? What kind of fluorescent ink was used in the experiment?  

Author Response

R) In this manuscript, the authors presented a compact, handheld, and cost-effective Fourier light field microscope (FLMic) for bio-sample imaging in both bright-field and fluorescent modes. The authors employed a M21 lens to replace infinity-corrected microscope objective, and used a field stop in illumination system to simplify the imaging system. Both the theoretical analysis and the experiment results have been given to demonstrate the performance of the proposed device. I think the proposed methods could be helpful in the applications of light field microscopy. Therefore, I suggest that the manuscript would be accepted if the authors make some modifications. Here are my comments:

A) First, we would like to thank the reviewer for his/her comments that helped us to improve the manuscript quality.

R) 1. In page 6, the authors discussed the spatial lateral resolution of the imaging system by imaging a USAF 1951 test target. They calculated the spatial lateral resolution by using the elemental images rather than the refocused images. I suggest the authors use the refocused images to calculate the spatial lateral resolution as the experiment results.

A) The resolution limit presented (see Eq. (1)) refers directly to the elementary images, because it only considers the reduction of the effective numerical aperture and does not consider the reconstruction process. Nevertheless, to show that the algorithm does not reduce the resolution, we have added a figure with the reconstructed image and its measured resolution, showing the agreement between the resolution of both the EIs and the reconstructed image.

R) 2. In page 8, the authors demonstrated the fluorescent imaging results. I suggest the authors provide some further details about experimental process. For example, did the authors use the optical filter? If so, how did the author mount the filter in the proposed system? What kind of fluorescent ink was used in the experiment?

A) As stated in the text, we used a reused colour plastic filters extracted from 3D movie glasses, keeping the low cost purpose of the system. The filter was placed between the sample and the mount of the M12 lens. The knot of the cotton fibres was dyed with Rhodamine 123. To provide more details about the fluorescence experiment, we included the spectral peak value and spectral width of both, the LED and the filter.